# Role of Nitric Oxide in Gene Expression Regulation during Cancer: Epigenetic Modifications and Non-Coding RNAs

**DOI:** 10.3390/ijms22126264

**Published:** 2021-06-10

**Authors:** Patricia de la Cruz-Ojeda, Rocío Flores-Campos, Sandra Dios-Barbeito, Elena Navarro-Villarán, Jordi Muntané

**Affiliations:** 1Institute of Biomedicine of Seville (IBiS), Hospital University “Virgen del Rocío”/CSIC/University of Seville, 41013 Seville, Spain; patricia.cruz.ojeda@gmail.com (P.d.l.C.-O.); rflores-ibis@us.es (R.F.-C.); sandra_dios89@hotmail.com (S.D.-B.); e.navarrovillaran@gmail.com (E.N.-V.); 2Networked Biomedical Research Center Hepatic and Digestive Diseases (CIBEREHD o Ciberehd), Institute of Health Carlos III, 28029 Madrid, Spain; 3Department of Medical Physiology and Biophysics, University of Seville, 41009 Seville, Spain; 4Department of General Surgery, Hospital University “Virgen del Rocío”/CSIC/University of Seville/IBiS, 41013 Seville, Spain

**Keywords:** hepatocarcinoma, miRNA, nitric oxide synthase, lncRNA, tumor-associated macrophages, cancer associated fibroblasts, cancer stem cells

## Abstract

Nitric oxide (NO) has been identified and described as a dual mediator in cancer according to dose-, time- and compartment-dependent NO generation. The present review addresses the different epigenetic mechanisms, such as histone modifications and non-coding RNAs (ncRNAs), miRNA and lncRNA, which regulate directly or indirectly nitric oxide synthase (NOS) expression and NO production, impacting all hallmarks of the oncogenic process. Among lncRNA, HEIH and UCA1 develop their oncogenic functions by inhibiting their target miRNAs and consequently reversing the inhibition of NOS and promoting tumor proliferation. The connection between miRNAs and NO is also involved in two important features in cancer, such as the tumor microenvironment that includes key cellular components such as tumor-associated macrophages (TAMs), cancer associated fibroblasts (CAFs) and cancer stem cells (CSCs).

## 1. Role of Nitric Oxide (NO) in Epigenetic Regulation during Cancer

### 1.1. NO and Nitrosative Stress

Nitric oxide (NO) has emerged during the last decades as a critical mediator of inter- and intracellular signaling pathways. In fact, NO has a dual role in cancer, showing pro- or anti-tumoral properties in a dose-, time- and compartment-dependent manner (Figure 1). Therefore, it might determine tumor progression, therapy efficacy and prognosis [1]. Endogenous NO free radical production involves the family of nitric oxide synthases (NOS), which catalyze the conversion of L-arginine and O_2_ into L-citrulline and NO, respectively. Cofactors and redox molecules, such as nicotinamide adenine dinucleotide phosphate (NADPH), flavin adenine dinucleotide (FAD), flavin adenine mononucleotide (FMN), tetrahydrobiopterin (BH4) and calmodulin, also participate in this reaction [2]. The following three NOS isoforms have been described: neuronal NOS (nNOS, NOS1), inducible NOS (iNOS, NOS2) and endothelial NOS (eNOS, NOS3) [3]. NOS1 and NOS3 are constitutive isoforms controlling vascular function in a Ca^2+^-dependent manner, while NOS2 is Ca^2+^-independent generating high amounts of NO.

Regarding its chemistry, NO is a small lipophilic molecule that rapidly diffuses through the cell membranes. NO is also very unstable in presence of O_2_, which deters its short-biological life. Moreover, given its nature as a free radical, NO reacts with the unpaired electrons of other molecules. For instance, the reaction of NO with O_2_ or superoxide ion gives rise to nitrogen dioxide (NO_2_) and peroxynitrite (ONOO^−^), respectively [4]. Furthermore, NO oxidation also produces dinitrogen trioxide (N_2_O_3_), dinitrogen tetroxide (N_2_O_4_), nitrate (NO_3_^−^), and nitrite (NO_2_^−^) [5]. NO mediates nitrosative stress through cyclic guanosine monophosphate (cGMP)-dependent or -independent mechanisms, through interactions with metals and thiol groups [6,7]. Further, cGMP-independent pathways have been earning interest during the last years, due to their ability to modify protein function through post-translational modifications (PTMs). Some of the best characterized NO-dependent posttranslational modifications are S-nitrosation, tyrosine nitration, S-nitrosoglutathione (GSNO) or oxidation towards sulfenic acids [8].

### 1.2. Nitric Oxide in Cancer Pathogenesis

NO regulates critical aspects of cancer induction and progression (Figure 1). In fact, patients with hepatocellular carcinoma (HCC) display high levels of NO derivates in serum and tumors associated with increased NOS2 and NOS3 expression [9]. In nasopharyngeal carcinoma cells, pro-apoptotic autophagy is inhibited by NOS1 overexpression through the activation of the mammalian target of rapamycin (mTOR) and upstream AKT signaling pathways by S-nitrosation of the phosphatase and tensin homolog (PTEN) [10]. In the tumor microenvironment, pro-tumorigenic cancer associated fibroblasts upregulate NOS1 together with nuclear factor erythroid 2-related factor 2 (NRF2) and hypoxia inducible factor 1α (HIF1α) [11]. In colon cancer, the mitochondrial localization of NOS1 contributes to apoptosis resistance and to the maintenance of low levels of reactive oxygen species [12]. On the other hand, given the versatility of NOS2, its deregulation has been implicated in tumor progression [13,14]. For instance, melanoma-derived exosomes promote the polarization of macrophages to an M1 (anti-tumoral profile) or M2 (pro-tumoral profile) phenotype, with increased upregulation of NOS2 and arginase, respectively, thereby suggesting a prooncogenic role for NO in the tumor microenvironment [15]. Other studies have characterized the role of NOS2 and NO in macrophages pro-inflammatory phenotype, being responsible for metabolic reprogramming and cytokine production [16]. In breast cancer, the inhibition of NOS2 with L-NMMA enhanced docetaxel-induced apoptosis in triple-negative breast cancer (TNBC) cellular and mouse models, which may indicate that high levels of NO contribute in these models to chemoresistance through endoplasmic reticulum (ER) stress-related pathways [17]. As a matter of fact, high levels of NOS2 have been found in metaplastic breast cancer, correlating with a poor clinical outcome [18]. Similarly, elevated levels of NOS2 in pancreatic ductal adenocarcinoma were related to lower survival rates, and NOS2 deficiency in genetically engineered mice led to reduced proliferation, migration and invasion [19]. In contrast, other studies have also shown that NOS2 expression was downregulated (mRNA and protein) in liver tumor samples compared to adjacent healthy liver tissues. Furthermore, NO production was lower in metastatic HCC, showing the importance of this free radical not only in liver carcinogenesis, but also in progression [20]. HCC tumor aggressiveness, relapse and chemoresistance have been associated with liver cancer stem cells (CSCs). In particular, NOS2 has been proposed as the driver of Notch1 signaling activation in CD24^+^CD133^+^ CSCs to maintain self-renewal and growth properties [21]. Intriguingly, the research carried out by Ikeguchi et al. [22] in HCC samples could not find any correlation of NOS2 expression and patient outcome, proliferative properties and apoptosis occurrence. Not less important, NOS3 has also been related to tumorigenic processes. NOS3 expression was found to be higher in total extracts from colon cancer biopsies compared to non-tumoral adjacent tissue. In contrast, NOS3 expression was reduced in the HCT116 colon cancer cell line in the same study. Olah et al. [23] have demonstrated that NO signaling is necessary for cell survival, being excessively high or low levels of nitrosative stress prejudicial for cell growth. The overexpression of NOS3 and concomitant NO mediate antiproliferative effects in hepatoblastoma cells through the modulation of the redox state of thioredoxin (Trx) and glutaredoxin (Grx) [24]. In this sense, Sorafenib as a tyrosine kinase inhibitor, as recommended treatment for patients in the advanced stage, reduces Trx1 expression and NOS3-dependent NO generation, which is related to the induction of apoptosis in liver cancer cells [25].

### 1.3. Introducing Epigenetic Regulation Induced by NO

Epigenetic regulation involves heritable and reversible changes in gene expression that do not involve DNA sequence alterations [26]. Most common epigenetic regulators include DNA methylation, histone post-translational modifications, such as acetylation, methylation, or phosphorylation, as well as non-coding RNAs (ncRNAs) involved in the altered pattern of mRNA translation [27]. The review describes recent findings deciphering the alteration of epigenetic regulation by NO in cancer (Table 1 and Table 2).

#### 1.3.1. NO and DNA Methylation

DNA methylation is an epigenetic modification in which a methyl group is covalently bound to the 5th carbon of the cytosine pyrimidine ring in a CpG dinucleotide [28]. The involved enzymatic reaction uses S-adenosyl-methionine as a methyl donor and is carried out by three separate DNA methyltransferases (DNMT1, DNMT3a, and DNMT3b) [29]. NO induces DNMTs posttranscriptional activity increase, resulting in the accumulation of CpG island methylation and the suppression of gene expression [30] (Figure 2A, Table 1). Indeed, there are the following two major altered methylation patterns observed in cancer: global DNA hypomethylation and promoter DNA hypermethylation [29]. DNA methylation regulates cancer by silencing tumor suppressor genes though hypermethylation or activating oncogenes through demethylation [31,32]. DNA methylation of promoter CpG islands of tumor suppressor genes such as *brca1*, *cdh1 (E-cadherin)*, *cdkn2a (p16)*, and *retinoblastoma* are known to be involved in a variety of cancer types [33]. The primary biological outcomes are the control of cell proliferation, gene expression and the mitotic G1/S transition [32]. Thus, the dysregulation of these hypermethylated genes has been associated with essential tumor properties such as tumor cell proliferation, anti-apoptosis, neo-angiogenesis, invasive behavior, and chemotherapy resistance [34].

Different studies demonstrated that CpG island hypermethylation occurs in the premalignant stages and tends to accumulate during multistep hepatocarcinogenesis [35,36]. Moreover, Lee et al. [35] suggested that the CpG island hypermethylation of *cox-2* or *p16* might be potential molecular markers for the identification of HCC, and also that the CpG island hypermethylation of *e-cadherin* or *gstp1* might be used as a potential biomarker for the prognostication of HCC.

Few studies address the impact of NO production or NOS expression and changes in DNA methylation patterns. COX2 activity is enhanced by NOS2-derived NO, which promotes angiogenesis and cell differentiation [37,38,39] and tumor growth, invasion and metastasis potential [40,41,42]. Hence, the assessment of the correlation between COX2 and NOS2 expression and microvessel density in HCV-positive HCCs suggested its importance in the pathogenesis of the disease [43]. In these settings, studies have been carried out to assess the function of NO in epigenetic modifications during carcinogenesis. NO has been suggested to play an important role in epigenetic modifications during infection-driven gastric cancer. *H. pylori* infection increases NO production in gastric cancer cells, leading to aberrant DNA methylation, both processes being reversed by a NOS inhibitor such as L-NAME administration [44]. In this sense, NOS2-derived NO, induced by *H. pylori*, causes silencing of tumor suppressor genes such as *runx3* or *runx2* by DNA methylation [45]. Also, interleukin-1β (IL-1β) induces *E-cadherin* methylation, leading to a decrease in E-cadherin expression at both mRNA and protein levels through NO during *H. pylori* infection, which links inflammation to carcinogenesis [44]. These findings suggest the involvement of NO in the activation of DNMT and a resulting altered DNA methylation pattern.

Deregulated genes by epigenetic silencing may cause ectopic expression of genes in cancer cells, which can lead to inflammation-associated cancers. Ectopic expression of activation-induced cytidine deaminase (AID) is known to be caused by NO. Hence, the study addresses whether NO modulates the AID expression and examines the implication of epigenetics deregulation in this ectopic expression. Tatemichi et al. [46] suggested that NO enhances AID and NOS2 expression in cancer cells involving CpG demethylation, resulting in greater frequencies of gene mutation.

#### 1.3.2. Histone Posttranslational Modifications in Cancer

The nucleosomes conform the fundamental unit of chromatin, and are made of a 147-base-pair segment of DNA around the four core histones (H3, H4, H2A and H2B). Histone tails contain high levels of lysine and arginine residues, which can be commonly modified by acetylation, methylation, phosphorylation, citrullination or ubiquitination [47]. Prominently, NO can alter cancer epigenetic regulation through acetylation and methylation of the core histone protein tails, and also through phosphorylation to control the DNA damage response [47,48] (Figure 2, Table 1).

##### Histone Acetylation

The acetylation of lysine residues neutralizes the positive charge of the histone tail, and is therefore generally associated with chromatin relaxation and transcriptional activation [49]. The acetylation level of histones is determined by the equilibrium between the activities of the following two groups of enzymes: histone acetyltransferases (HATs) and histone deacetylases (HDACs) [46]. The main physiological functions of these enzymes are to maintain the steady-state levels of the lysine acetylation of histone and non-histone proteins, regulating chromatin condensation and relaxation balance. It plays a relevant role in tumor cell proliferation, metastasis, angiogenesis, resistance to apoptosis and alteration of the cell cycle, among others [50]. However, although the mechanisms of HDACs action in cancer are diverse and some of them remain unknown, the aberrant expression of HDACs in tumors is widely associated with silencing tumor suppressor genes transcription or upregulation of oncogenes [51,52,53,54], overall associated with poor outcomes in patients. Jung et al. [55] demonstrated HDAC8 overexpression in HCC and exerting its knockdown antioncogenic effects, possibly due to the high expression of p53 and the acetylation of p53. In addition, NO is also involved in increased AID expression by HDAC inhibition [46].

HDACs function can be modulated by nitrosative stress at multiple levels. Different studies in neurons or C2C12 myoblasts showed a decrease in HDAC2 activity by S-nitrosation, leading to gene activation [56,57,58]. In addition, it has been proved that the knockdown of HDAC1, HDAC2 or HDAC3 can promote the development of HCC [59]. Moreover, several class II HDACs as SIRT1 can be inactivated due to S-nitrosation, as well as SIRT6, making these inhibitions possibly oncogenic. Recent studies have addressed HDAC6 as a target of NO. They found that NO induces HDAC6 S-nitrosation by exposing epithelial cells to physiological NO donors, as well as endogenously by NO produced from a mixture of inflammatory cytokines, such as TNF-α (tumor necrosis factor α), IL-1β and IFN-γ (interferon-γ), which stimulate NOS2 expression [60]. Moreover, SNOC (S-nitrosoglutathione-oligosaccharide-chitosan), a NO donor, treatment significantly attenuated HDAC6 activity, indicating that NO directly inhibits its enzymatic activity by S-nitrosation (Figure 2B, Table 1). They concluded that HDAC6 is regulated by redox state, and excess amounts of NO may induce pathophysiological conditions, which increase the accumulation of misfolded proteins and subsequent cell death [60]. Histone deacetylase inhibitors and NO play a relevant role against the progression of muscular dystrophy in MDX mice. In this sense, Colussi et al. [58] showed that HDAC2 was up-regulated in dystrophic muscles, and its S-nitrosation by NO donors weakens its enzymatic function, highlighting the potential therapeutic role of HDAC inhibitors and NO donors in Duchenne muscular dystrophy [58].

Recent studies have discovered an important role of the histone acetylation on different proteins that are considered specific biomarkers for HCC. In this sense, alpha-fetoprotein (AFP) can be either acetylated or deacetylated. Xue et al. [61] demonstrated that AFP interacts with and is regulated by CREB-binding protein, or CBP (acetyltransferase) and SIRT1 (deacetylase) in HCC cells. Also, CBP is involved in the apoptotic pathway in several tumor cells. In addition, CBP silencing decreased NO production by downregulating the NOS3 expression. Furthermore, the increased apoptosis of endothelial cells coincided with a reduction in NO, and it was reversed by NO donors. Hence, the research demonstrated the association of CBP silencing with the low expression of NOS3 and NO production, and the increase in endothelial cells apoptosis [62].

##### Histone Methylation

Histone mono-, di-, or tri-methylation can take place in both the lysine and arginine residues of histones. Given that multiple methylation states exist for both aminoacidic residues, a great network of complexity arises with different methylation profiles [47]. Lysines 4, 9, 27, 36 and 79 are targeted in histone 3 for methylation. Typically, methylation at K4, K36 or K79 are associated with transcriptional activation, whereas methylation at K9 or K27 have been linked with gene repression. Lysine 4 in histone 4 can be also methylated. The enzymes involved in these epigenetic marks are known as methylases, and the erasers of methyl groups are named demethylases [63].

NO has been shown to alter the methylation status by inhibiting histone 3 and lysine 9 demethylase KDM3A (Jumonji domain containing 1A) activity and led to an accumulation of the histone H3K9me2 substrate [64] (Figure 2C). To compensate this inhibition, the demethylases KDM3B, KDM4A, KDM4B, KDM4C, and KDM4D were found to be up-regulated, especially KDM1 and KDM7A. Functional studies suggest that NO acts directly into the catalytic site of the demethylase, forming a nitrosyl–iron complex [64]. KDM3A mRNA and protein expressions are significantly increased in HCC-derived tumors compared with non-cancerous tissues, which correlated to reduced disease-free survival and increased tumor recurrence rates [65]. Furthermore, in vitro experiments have shown that KDM3A downregulation by siRNA decreased cell proliferation, invasion and epithelial-to-mesenchymal transition induced by hypoxia [65]. One of the targets that KDM3A regulates is the hypoxic factor adrenomedullin. During hypoxia, KDM3A demethylates H3K9me2 and allows chromatic relaxation for adrenomedullin transcription, which enhances the proliferative properties of liver cancer cells. KDM3A depletion suppresses liver cancer cells tumorigenicity in nude xenograft mice models [66]. All in all, these results might suggest that a decrease in demethylase activity by NO is related to anti-tumorigenic roles, as seen by the decreased proliferation in vitro, reduced tumor growth in vivo and tumor recurrence.

Regarding H3K9 histone methyltransferases (HMTs), NO has been shown to downregulate G9a together with KDM3A demethylase to maintain the levels of histone H3K9me2 [64]. Snail-2 transcription factor interacts with G9a H3K9 methylase and histone deacetylases to repress the promoter of the epithelial marker E-cadherin. Therefore, Snail-2 upregulation promotes epithelial mesenchymal transition (EMT) and increased aggressiveness in HCC [67]. G9a is highly expressed in human HCCs and significantly associates with portal vein invasion or tumor microsatellite formation. This methyltransferase might exert its pro-tumorigenic functions by epigenetic repression of tumor suppressor genes such as RARRES3 [68]. Therefore, downregulation of G9a, induced by NO, could be treated as a potential target for a therapeutic response.

SET domain bifurcated 2 (SETDB2), the suppressor of variegation 3–9 homolog 2 (SUV39H2) trimethylating enzymes and PRDM2 methylase were found to be up-regulated in response to NO [64]. A model of risk has been constructed according to the expression data of methyltransferase-like protein 6 (METTL6), RNA polymerase III subunit G (POLR3G), phosphoribosyl pyrophosphate amidotransferase (PPAT), SETDB2 and SUV39H2 in 352 HCC patients, obtained from The Cancer Genome Atlas Liver Hepatocellular Carcinoma database. This model was able to predict overall survival and disease progression. Moreover, the high-risk cases were related to cell proliferation, MYC targets and DNA repair, a higher p53 mutation rate as well as a protumoral immune microenvironment [69]. Similarly, a high expression of histone lysine methyltransferase, a suppressor of variegation 3–9 homolog 1 (SUV30H1), correlates with HCC progression, being responsible for migration and metastasis [70]. The role of this transferase has been confirmed in colorectal and breast cancer, proposing H3K9 aberrant trimethylation due to increased SUV39H1 expression as the driver mechanism for migration [71]. Although not demonstrated in cancer, NO has been shown to indirectly regulate SUV39H1. In the presence of nitrosative stress, S-nitrosation of GADPH occurs, which is necessary for its binding to Siah ubiquitin ligase. Under these circumstances, Siah targets SUV39H1 for proteasomal degradation and consequently reduces the H3K9 trimethylation status [72] (Table 1).

Histone H3K4 methyltransferase *MLL* gene is commonly rearranged and translocated in some pediatric leukemia, such as B-cell precursor acute lymphoblastic leukemia (BCP-ALL). In cellular models of BCP-ALL with gene rearrangements, AMP-activated protein kinase (AMPK) signaling pathways are hyperactivated to promote cell survival. The inhibition of AMPK with compound-C alone induces cell cycle alterations and apoptosis through the mitochondrial pathway, and also synergizes with chemotherapeutic agents [73]. In HCC, hepatocyte growth factor (HGF)–MET signaling promotes the DNA binding factor ETS2 to interact with MLL. This interaction targets MLL to the promoter of matrix metalloproteinase 1 (MMP1) and 3 (MMP3), thereby inducing cell invasion [74]. Little has been described about the connection of the NO and H3K4 methylation status. In models of inflammatory diseases, such as osteoarthritis, IL-1 stimulates the expression of NOS2 and COX2 by increasing H3K4 di and trimethylation at their promoters. SET-1A, but not MLL methyltransferase, is involved in H3K4 methylation at the NOS2 and COX2 promoters [75]. KDM5A-D enzymes demethylate di and trimethylated H3K4. A high expression of KDM5A is a requisite for treatment resistance in breast and lung cancer cells [76]. Similarly, increased levels of KDM5B are found in HCC tumor samples and cellular models. A high expression of KDM5B in clinical samples correlates with a lower differentiation status, tumor size and TNM stage [77]. Mechanistically, KDM5B promotes EMT by decreasing the levels of epithelial markers E-cadherin and α-catenin, and increasing the levels of mesenchymal markers N-cadherin and vimentin. This demethylase exerts its protumoral properties through the regulation of H3K4me3 at the PTEN promoter [78]. Nonetheless, the connections between cancer, NO and H3K4 demethylases require additional research (Table 1).

Contrary to H3K4 di or trimethylation, the methylation of histone H3K27 triggers gene suppression. Transcriptional inhibition mediated by H3K27me3 is necessary for the plasticity of hepatic cells, which might be important for cancer progression. The inhibition of demethylation with compound GSK-J4 leads to increased trimethylation status, reduced acetylation and decreased hepatocyte markers, such as albumin and Cyp3A4. Also, during the differentiation process, demethylation is favored by decreased levels of EZH2 methyltransferase [79]. In addition, EZH2 has been shown to be involved in NOS2 expression in ChIP-on-chip experiments, controlling H3K27me3 around the transcription start site. The inducibility of NOS2 was studied in EZH2-deficient endothelial cells. Upon stimuli with interferon γ, TNFα, IL-1β or lipopolysaccharide, NOS2 was not induced in this study, suggesting other regulatory mechanisms besides EZH2. Nonetheless, no alternative methyltransferases or demethylases were explored [80].

A major issue in tumor aggressiveness is related to CSCs maintenance through epigenetic regulation. One of the best-known transcription factors involved in stem cell renewal is Oct4. NO promotes Oct4 expression (Figure 2C). In the absence of nitrosative stress stimuli, Oct4 forms a complex with caveolin-1, which mediates ubiquitin-mediated proteasomal degradation. However, upon nitrosative stress stimuli, Akt1 phosphorylates caveolin-1 and disrupts Oct4 degradation [81]. In HCC, the expression of Oct-4 and CSCs maintenance are associated with increased H3K36 methylation. Liver CSCs down-regulate the expression of the transcription factor ZHX2, which controls the expression of the H3K6 demethylase KDM2A. Therefore, increased methylation status of stemness markers, such as Oct4, is required for CSC phenotype [82]. It would be necessary to unravel the exact role of NO in the epigenetic marks of Oct4 in HCC.

**Table 1 ijms-22-06264-t001:** Crosstalk between NO and DNA methyltransferases, histone deacetylases, histone methyltransferases and histone demethylases. The table indicates the names of the enzyme involved and its substrate. It summarizes the connections between epigenetic regulators and NO and the impact in cancer.

Epigenetic Regulation	Enzyme	Transcriptional Role	Crosstalk between NO and Epigenetic Regulators	Impact of the Regulatory Mechanism in Carcinogenesis	References
DNA methylation	DNMT not specified	Transcriptional repression	NOS-2-derived NO reduces tumor suppression genes expression	Pro-tumoral	[45]
	DNMT not specified	Transcriptional repression	NO induces E-cadherin methylation by IL-1B decreasing E-cadherin expression	Pro-tumoral	[44]
	DNMT not specified	Ectopic expression	NO causes ectopic expression of AID and enhances NOS2 expression	Pro-tumoral	[46]
Histone deacetylation	HDAC6	Transcriptional repression	NO induces HDAC6 S-nitrosation	Pro-tumoral	[60]
	HDAC2	Transcriptional repression	NO S-nitrosation weakens HDAC2 enzymatic function	Anti-tumoral	[58]
	CBP	Transcriptional repression	CBP silencing decreases NO production by downregulation NOS-3	Anti-tumoral	[61,62]
	SIRT1				
Histone methylation	G9a	Transcriptional repression	NO downregulates expression	Anti-tumoral	[64,67,68]
	SETDB2		NO upregulates expression	Pro-tumoral	[64,69]
	SUV39H2				
	SUV30H1		NO indirectly targets SUV20H1 for proteasomal degradation	Anti-tumoral	[70,71,72]
	MLL	Transcriptional activation	Not described	Pro-tumoral	[74]
	SET-1A		SET-1A trimethylates NOS2 promoter in response to IL-1	Pro-tumoral	[75]
	EZH2	Transcriptional repression	EZH2 does not control NOS2 expression. Other mechanism should be involved	Pro-tumoral	[79,80]
Histone demethylation	KDM3A	Transcriptional activation	NO inhibits KDM3A by forming a nitrosyl–iron complex	Anti-tumoral	[64,65,66]
	KDM3B		NO upregulates expression. Compensatory mechanism in response to NO mediated KDM3A inhibition	Not described	[64]
	KDM4A				
	KDM4B				
	KDM4C				
	KDM4D				
	KDM1				
	KDM7A				
	KDMA	Transcriptional repression	Not described	Pro-tumoral	[76]
	KDMB				[77,78]
	KDM2A	Transcriptional repression	NO promotes the expression of Oct-4, which is related to reduced expression of demethylase KDM2A	Pro-tumoral	[81,82]

##### Histone Phosphorylation

Histone phosphorylation in response to nitrosative stress in cancer research has also been described. The histone variant H2AX can be phosphorylated on Ser139 (γH2AX) by phosphoinositide 3-kinase-related protein kinases (PIKKs) ATM-12, ATR-13 and/or DNA-dependent protein kinase (DNA-PK) in response to DNA double-strand breaks (DSBs). It takes place one megabase around the DSB, thereby providing a useful reporter of DNA damage, commonly used in immunofluorescence and flow cytometry techniques [83]. NO donors, such as NO-releasing acetylsalicylic acid (NO-ASA), have shown anti-tumoral properties. The exposure of human B-lymphoblastoid TK6 cells to NO-ASA induced H2AX phosphorylation in a dose-dependent manner, being this phosphorylation-specific to S-phase, and caspase-3 activation [84] (Figure 2D). Thus, NO-ASA might be used as a promising genotoxic agent for highly aggressive proliferating tumors [84]. Similarly, NO and hydrogen sulfide (H_2_S) releasing aspirin (NOSH-ASA) has also shown promising anti-cancer properties in vitro and in xenograft mouse models [85]. In cellular models of pancreatic cancer, NOSH-ASA blocks cell cycle progression and induces apoptosis through caspase-3 activation and oxidative stress. In vivo, NOSH-ASA reduces tumor volume and mass up to 90% and 75%, respectively, compared to vehicle-treated mice. Tumor reduction was mediated by reduced proliferation and increased apoptosis by TUNEL. Regarding signaling pathways, NOSH-ASA increased oxidative stress, p53 and NOS2 expression, and downregulated NF-κB and FoxM1 [86].

Also, NO could act as a driver of genomic instability in carcinogenesis. Barrett’s esophagus (BE) is a metaplastic condition caused by chronic gastroduodenal–esophageal reflux, constituting a high risk of esophageal adenocarcinoma cells (EACs) development. During reflux, huge concentrations of NO are present in the esophageal lumen, where NO derivatives, such as ONOO^−^ or N_2_O_3_, are thought to mediate DNA damage. In vitro, NO and HNO_3_ are able to cause DSBs during S-phase in nondysplastic, high-grade dysplasia, and adenocarcinoma cell lines [87]. In HCC, phosphorylation of H2AX could also be taken as a driver mechanism for progression. Under hypoxia, tumor cells could develop DSBs, which in turn lead to the expression of γH2AX. Epidermal growth factor receptor (EGFR) might translocate into the cell nucleus and combine with γH2AX. After that, HIF-1α and vascular endothelial growth factor (VEGF) expression occurs to promote angiogenesis. Consequently, elevated levels of γH2AX, HIF-1α and VEGF in the serum of patients with HCC submitted to liver transplantation constitute a biomarker of poor prognosis [88].

#### 1.3.3. Non-Coding RNAs

##### Small RNAs

Small RNAs cover the following two main subsets of non-coding RNAs (ncRNAs): housekeeping and regulatory ncRNAs. The first classification involves ribosomal RNAs (rRNAs), transfer RNAs (tRNAs), small nuclear RNAs (snRNAs), small nucleolar RNAs (snoRNAs) and telomerase RNAs, whereas regulatory ncRNAs comprise microRNAs (miRNAs), small interfering RNAs (siRNAs), and piwi-interacting RNAs (piRNAs) [89,90]. miRNAs are the most abundant small ncRNAs (18–24 bp) that regulate the expression of target mRNAs, thereby contributing to the epigenetic regulation of gene expression, and their expression is altered in cancer (Table 2). miRNAs have been linked to NO signaling, being the cause or consequence of NO dysregulation during cancer. It has been stablished that NO and p53 are able to regulate miRNA expression and the development of lymphomas [91]. In this context, the experimental downregulation of p53 and NOS2 expression using KO mice reduces the expression of miR-34b/c and miR-29b/c, respectively [92] (Figure 3A1). Furthermore, miR-29b has been shown to downregulate PTEN, leading to increased cell migration and invasiveness potential in metastatic breast cancer [93] (Figure 3A1). In addition, miR-29b/c, whose expression could also be modulated by hypermethylation of its promoter, regulates DNMT3A expression, suggesting a potentially relevant crosstalk between both of the epigenetic modulators in gastric cancer [94] (Figure 3A1, Table 2).

In prostate cancer, NOS3 has been confirmed to be a shared target of miR-335 and miR-543. In this sense, the overexpression of miR-335 and miR-543 reduces NOS3 expression, and cell migration and invasiveness in cultured PC-3 cancer cells, being this connection confirmed in patients with metastatic prostate cancer [95] (Figure 3A2). In line with this, the transfer of miR-335-5p from stellate cells to liver cancer cells through exosomes reduces their proliferative and invasion potential both in vitro and in vivo [96]. The upregulation of miR-193b exerts antitumoral properties through reducing NOS2 activity in breast cancer (Figure 3A3). The described molecular mechanism suggests that the downregulation of miR-193b increases the expression of dimethylarginine dimethylaminohydrolase 1 (DDAH1), which negatively impacts the expression of the NOS inhibitor asymmetric dimethylarginine (ADMA), resulting in an increase in tumor severity by increasing proliferation and migration in breast cancer [97] (Figure 3A3, Table 2).

The interaction between miRNA and NO has also been investigated in the tumor microenvironment. In this sense, miR-193 and miR-30 appear to reduce transforming growth factor β (TGF-β)-dependent extracellular matrix accumulation in hepatic stellate cells in liver fibrosis [98]. Activated anti-tumoral M1 macrophages show an increased nitrosative state that coincides with increased miR-16 expression, which appears to be a requirement for turning macrophages from basal or M2 to M1 polarized states. Interestingly, miR-16 downregulates PDL-1 expression and consequently benefits CD4^+^ T cell-dependent antitumoral properties [99] (Figure 3A4). EACs display low levels of miR-155, which is associated with increased fibroblast growth factor 2 (FGF2) expression, cell proliferation, migration and invasiveness potential [100]. In addition, the overexpression of miR-155 exerts antitumoral properties, characterized by increased expression of TNF-α (tumor necrosis factor α), IL-12 and NOS2 expression, as well as a reduction in IL-10, arginase-1 and IL-22 in conditioned culture medium from tumor-associated macrophages (TAMs) [100] (Figure 3A5). Therefore, in this setting, high levels of miR-155 correlate with high NOS2 in TAMs, and reduced FGF2 expression in EACS overall diminish cancer cell proliferation [100] (Table 2). Recently, nRNA/snoRNA-derived nuclear RNA 3 has been pointed out as the molecular mechanism underlying NOS2 gene-specific targeting in macrophages. In resting macrophages, sdnRNA-3 participates in the formation of a closed chromatin domain of the *Nos2* promoter recruiting the antagonist chromatin regulator Mi-2β and increasing the H3K27me3 levels. High expression levels of sdnRNA-3 contributed to the pro-tumorigenic properties of M2 TAMs by decreasing NOS2 expression [101].

The exogenous administration of the NO donors S-nitroso-N-acetylpenicillamine (SNAP) and sodium nitroprusside (SNP) increased the expression of miR-155, while low endogenous NO generation decreased its expression through cGMP-dependent pathways in HepG2 cells [102]. Nonetheless, no further analyses were performed in this study to assess the impact of miR-155 in Hep3b aggressiveness. A recent study has demonstrated that miR-155 plays a pro-tumoral role in hepatocarcinogenesis through the inhibition of H3F3A expression and H3K27 methylation, which blocks the expression of the P21WAF1/CIP1 tumor suppressor gene [103]. Similarly, the downregulation of miR-122 has been found to be related to Sorafenib resistance in liver cancer cells [104]. The molecular mechanism underlying the proliferative properties of miR-122 silencing involves increased expression of its target SLC7A1, an arginine transporter, which provides an arginine substrate for NO production by NOS2 and increases cell proliferation in Sorafenib-treated Huh7 cells [104]. The administration of PD407824 or Ellipticine, which up-regulate miR-122 expression, could provide chemosensitivity in HCC [104]. Herein, these results support high levels of NO as an oncogenic driver in HCC, and support its downregulation for therapy effectiveness.

##### Long Non-Coding RNAs

Long non-coding RNAs (lncRNAs) are ncRNAs with more than 200 nucleotides. LncRNAs have diverse functions, such as in chromatin modification, transcription and post-transcriptional processing. LncRNAs also participate in many important cellular signal transduction regulations through epigenetic silencing, mRNA splicing, lncRNA–miRNA interactions, lncRNA–protein interactions and lncRNA–mRNA interactions [105,106]. LncRNAs regulate a wide variety of biological processes relevant to liver homeostasis and carcinogenesis [28] (Table 2). Among them are found HOX transcript antisense intergenic RNA (HOTAIR), HCC upregulated EZH2-associated lncRNA (HEIH), GABPB1-AS1 or urothelial carcinoma-associated 1 (UCA1) [107].

The role of lncRNAs is context-dependent and they might function as oncogenes or tumor suppressors [108]. For example, HOTAIR is known by its epigenetic role in chromatin structure modification as a modular scaffold for histone modification complexes [109]. The role of HOTAIR in the development and progression of cancer has been described in breast cancer [110] and HCC [111]. In order to study the molecular mechanisms of lncRNAs as chromatin modifiers that affect transcription in a hormone-dependent and -independent fashion, the molecular interactions of HOTAIR and MALAT1 with estrogen receptor α (ERα)/estrogen receptor β (Erβ) in prostate or breast cancer cells, respectively, have been studied.

It was observed that ER/NOS3 interacting with MALAT1 and co-transcriptional repressor generate a complex resulting in closed chromatin conformation in the absence of estradiol. However, the administration of estrogens promotes MALAT1 and co-repressor detachment from the ER/NOS3 complex, which recruits HOTAIR and initiates transcription in estrogen-target promoters [112].

HEIH has been described as an oncogenic lncRNA in HCC [113] and TNBC [114]. In close relation with these studies, Guo et al. [115] have previously shown that miR-939-5p downregulates NOS2 expression in cultured human hepatocytes. The connection among all elements suggests that HEIH might play a relevant role in these settings. In fact, HEIH reduces miR-939-5p expression, which is associated with the upregulation of NOS2-dependent NO generation and tumor promotion in TNBC [114] (Figure 3B1, Table 2).

The expression of UCA1 has been demonstrated in HCC [116], and ovarian cancer and breast cancer [117]. The reduction in miR-204 has been related to UCA1 expression in acute myeloid leukemia (AML) cultured cells and in patients. The proapoptotic and antiproliferative properties of miR-204 were associated with the reduced expression of SIRT1, COX2 and NOS2 in AML cells [118] (Figure 3B2). In this setting, UCA1 exerted sponging interaction with miR-204 and prevented all downstream events in AML cells [118]. In summary, these results suggest that the UCA1/miR-204/SIRT1/NOS2/COX2 axis regulates cell proliferation and apoptosis in AML cells [118] (Table 2).

H19 has been shown to downregulate miR-148b-3p, which participates in tumor growth, proliferation and angiogenesis in different cancer models [119]. In liver pathophysiology, H19 plays a relevant role in hypoxic stress, reducing miR-148b expression in hepatic sinusoidal endothelial cells (HSEC) (Figure 3B3). Zhu et al. [120] showed that lncRNA H19 negatively regulated miR-148b-3p, which in turn was upregulating NOS3/NO and downregulating NOX4 in HSEC [120] (Table 2).

**Table 2 ijms-22-06264-t002:** Expression of miRNA and lncRNA in control and cancer. Low and high expression are indicated by “−” or “+”, respectively. Mechanisms linking cancer and NO are also specified.

		Type of Cancer	Expression		Molecular Mechanism	Interaction with NO	Impact of the Regulatory Mechanism in Carcinogenesis		References
			Control	Cancer					
**miRNAs**	miR-29b/c	Gastric cancer	−	+	Expression of miR-29b/c is regulated by NOS2	Not specified	NOS2↑–miR-29b/c↑–PTEN↓-Migration↑–Apoptosis↓	NOS2 regulates the expression of miR-29b/c, which in turns reduces PTEN and apoptosis, and increases migration	[92,93,94]
	miR-335, miR-543	Prostate cancer/ Liver cancer	+	−	Post-transcriptional regulation of NOS3	NOS3 mRNA degradation (miRNA target)	miR-335, miR-543↓–NOS3↑–Metastatic potential↑	miR-335 and miR-543 target NOS3 mRNA for degradation. In cancer, downregulation of these miRNAs, increases NOS3 expression leading to higher metastatic potential	[95,96]
	miR-193b	Breast cancer	+	−	Post-transcriptional regulation of NOS2 regulator DDHA1	DDHA1 mRNA degradation (miRNA target)	miR-193↓–DDAH1↑–ADMA↓–NOS2↑–Angiogenesis↑	Downregulation of miR-193b reduces DDAH1 mRNA degradation, which increases ADMA elimination and consequent increased NOS2 activity. This leads to increased angiogenesis	[97]
	miR-16	Pan-cancer (macrophages)	+	−	NO production	Not specified	miR-16↓–NO production↓–Pro-tumoral microenvironment↑ and miR-16↓–PD-L1↑–Pro-tumoral microenvironment↑	miR-16 in M1 macrophages is able to increase NO production, leading to an anti-tumoral microenvironment. Also, miR-16 targets PD-L1 mRNA for degradation, leading to reduced immunosuppression. In M2 macrophages, downregulation of miR-16 coincides in reduced NO production	[99]
	miR-155	Pan-cancer (macrophages)	+	−	Post-transcriptional regulation of NOS2	Not specified	miR-155↓–NOS2↓–FGF2↑–Proliferation↑	Downregulation of miR-155 decreases NOS2 expression and increases FGF2, promoting tumor proliferation	[100]
	miR-155	Liver cancer	−	+	Exogenous NO increases miR-155 expression	Not specified	miR-155↑–tumor suppressor gene P21WAF/CIP1↓	In liver cancer, upregulation of miR-155 by exogenous NO donors, blocks tumor suppressor gene P21WAF/CIP1	[102,103]
	miR-204	Acute myeloid leukemia	+	−	Post-transcriptional regulation of SIRT1, NOS2 and COX2	Not specified	miR-204↑–SIRT1↓/NOS2↓/COX2↓	In AML cells, miR-204 reduces expression of SIRT1, COX2 and NOS2 exerting proapoptotic and antiproliferative properties	[118]
	miR-939-5p	Triple-negative breast cancer	+	−	Post-transcriptional regulation of NOS2	Not specified	miR-939-5p↑–NOS2↑–NO↑	miR-939-5p downregulates NOS2 expression in cultured human hepatocytes and in TNBC	[119]
	miR-148b-3p	Liver cancer (Hepatic sinusoidal endothelial cells)	+	−	Post-transcriptional regulation of NOS3 and NOX4	NOX4 mRNA degradation (miRNA target)	miR-148b-3p↑–NOS3↑/NO↑–NOX4↓	miR-148b-3p regulates negatively NOX4, it also enhances NOS3 expression and NO production in HSEC	[120]
	miR-122	Liver cancer	+	−	Post-transcriptional regulation of SLC7A1 arginine transporter	SLC7A1 mRNA degradation (miRNA target)	miR-122↓–SLC7A1↑–Arginine↑–NO production↑–Cell proliferation↑	Downregulation of miR-122 promotes cell proliferation in liver cancer through upregulation of NO production. In particular, miR-122 targets arginine transporter SLC7A1. Under circumstances of reduced expression of miR-122, SLC7A1 is not degraded and arginine availability increases	[104]
**lncRNAs**	UCA1	Acute myeloid leukemia	−	+	Post-transcriptional regulation	miR-204 mRNA degradation	UCA1↑–miR-204↓–SIRT1↑/NOS2↑/COX2↑	UCA1 downregulates miR-204 expression and it enhances expression of SIRT1, NOS2 and COX2	[118]
	HEIH	Triple-negative breast cancer	−	+	Post-transcriptional regulation	miR-939-5p degradation	HEIH↑–miR-939-5p↓–NOS2↑–NO↑	In TNBC HEIH decreases miR-939-5p expression, which consequently enhances NOS2 expression and NO production	[119]
	H19	Liver cancer (Hepatic sinusoidal endothelial cells)	−	+	Post-transcriptional regulation	miR-148b-3p degradation	H19↑–miR-148b-3p ↓–NOS3↓/NO↓–NOX4↑	H19 negatively regulates miR-148b-3p, so it turns to downregulate NOS3/NO and upregulates its direct target NOX4 in HSEC	[120]

## 2. Concluding Remarks

NO has been identified and described as a dual mediator in cancer, being able to exert antitumoral and oncogenic properties according to dose-, time- and compartment-dependent NO generation. In addition, its impact is affected by the genetic background of the cell, hypoxia/re-oxygenation status and the presence of additional free radicals or scavengers. The moderated upregulation of NOS expression is widely associated with carcinogenesis, tumor progression and treatment resistance. As we have discussed above, NO influences different epigenetic regulators involving histones modifications and structural DNA binding proteins. Also, NO is able to dysregulate DNA methylation and acetylation, and promotes gene expression, inflammation, genomic instability and carcinogenesis.

Different studies suggest a close connection between NO and the expression of miRNAs. In addition, lncRNAs play a widely sponging role on miRNA expression, preventing their downstream events. In this sense, HEIH and UCA1 develop their oncogenic functions by inhibiting their target miRNAs, and consequently reversing the inhibition of NOS and promoting tumor proliferation. The connection between miRNAs and NO is also involved in two important features in cancer, such as the tumor microenvironment and the maintenance and renewal of CSCs. Here, we provide some clues about the role of increased NO in the transition among M1 and M2 TAMs, CAFs activation, and CSCs maintenance.

In conclusion, numerous epigenetic features previously related to cancer progression appear nowadays mediated by the alteration in NOS expression and NO production in cancer. More studies will decipher more molecular links between the epigenetic regulation of NOS and cancer, which might be relevant in order to increase the effectiveness of the treatment.

## Figures and Tables

**Figure 1 ijms-22-06264-f001:**
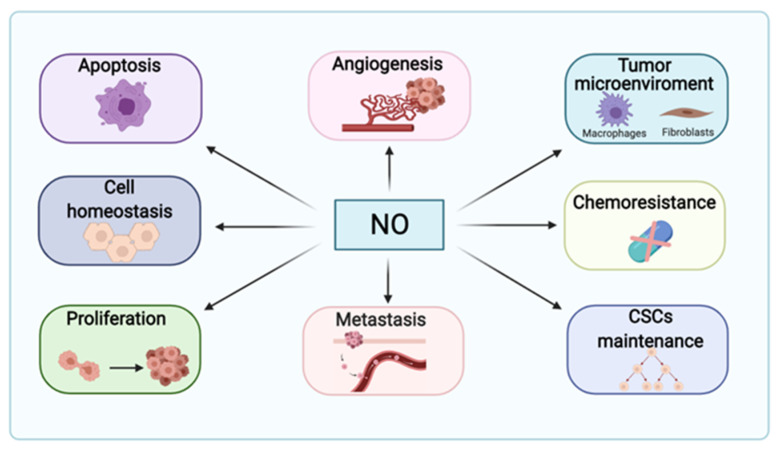
Implication of nitric oxide (NO) during carcinogenesis. Although NO is relevant for homeostasis, it may influence cell proliferation, metastasis potential, cancer stem cells (CSCs) maintenance and renewal, chemo- and apoptosis resistance, as well as modulating the tumor microenvironment and angiogenesis according to its concentration-, time- and compartment-dependent generation.

**Figure 2 ijms-22-06264-f002:**
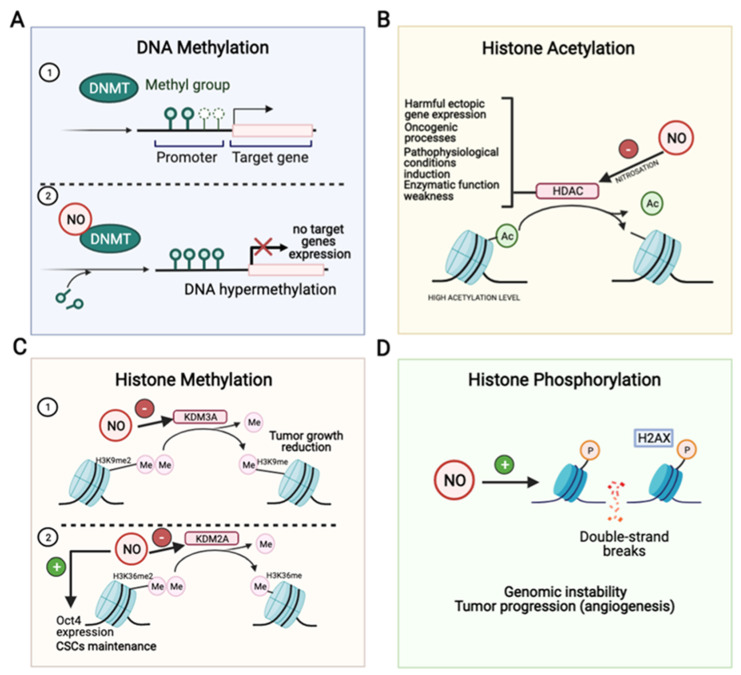
Impact of NO in DNA methylation (**A**), histone methylation (**B**), histone acetylation (**C**) and histone phosphorylation (**D**). DNA methyltransferases (DNMT) enzymes are responsible for methylating DNA cytosine residues. Genes with low promoter cytosine methylation are expressed (**A1**), but upregulation of DNMT protein expression and activity by NO leads to increased DNA methylation at promoter regions and repression of downstream associated targets (**A2**). NO inhibits histone deacetylases (HDAC) by S-nitrosation increasing acetylation level causing harmful ectopic gene expression, oncogenic processes, pathophysiological conditions induction and enzymatic function weakness (**B**). NO inhibits H3K9me2 lysine demethylase 3A (KDM3A) leading to decreased histone methylation status and tumor growth (**C1**). Nonetheless, NO promotes Oct4 expression and CSCs maintenance through inhibiting H3K36me2 demethylase KDM2A (**C2**). NO induces genomic DNA double-strand breaks and tumor progression (**D**). Acetylation, Ac; lysine demethylase 2A, KDM2A; methylation, Me; phosphorylation, P.

**Figure 3 ijms-22-06264-f003:**
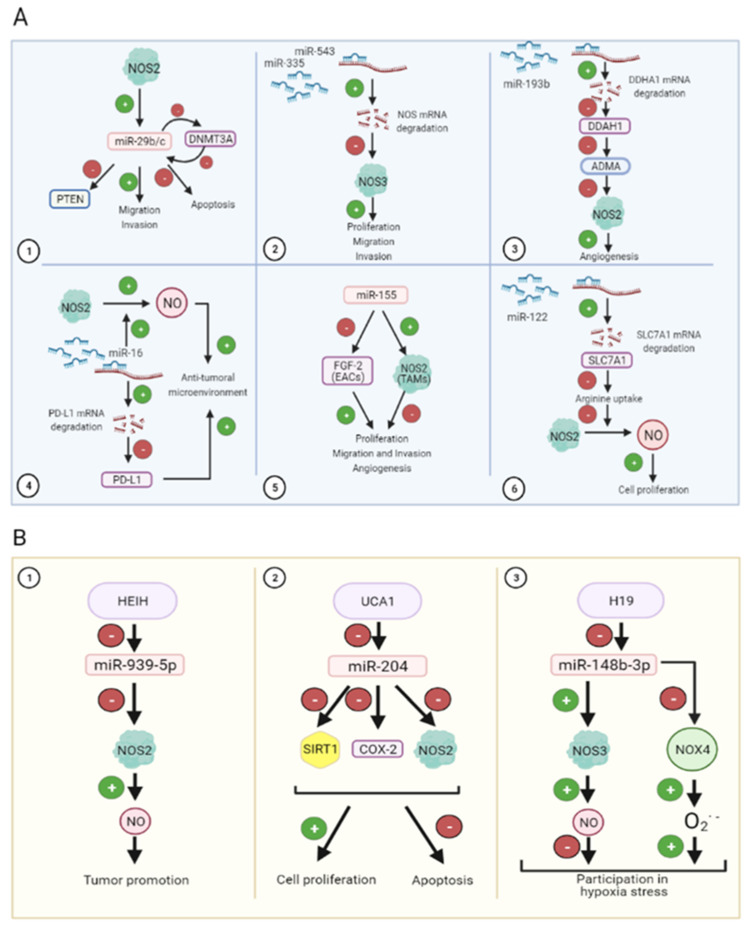
Implications of micro RNAs (**A**) and long non-coding RNAs (**B**) in carcinogenesis induced by NO. Inducible NOS (NOS2) positively regulates migration and invasion in cancer cells and negatively regulates phosphatase and tensin homolog (PTEN) suppressor gene and apoptosis through microRNA miR-29b/c expression induction. Furthermore, miR-29b/c represses DNA demethylase DNMT3A in a negative feedback loop (**A1**). Endothelial NOS (NOS3) also promotes proliferation, migration and invasion, which constitutes a shared target of miR-335 and miR-543 (**A2**). Activity of NOS2 is also controlled by miRNA expression. In particular, miR-193b targets dimethylarginine dimethylaminohydrolase 1 (DDAH1) enzyme, which removes asymmetric dimethylarginine (ADMA), a nitric oxide synthase (NOS) inhibitor (**A3**). miR-16 promotes NOS2 activity, increasing NO production, necessary for maintaining an anti-tumoral microenvironment. Moreover, miR-16 targets PD-L1, reducing immunosuppression (**A4**). miR-155 controls proliferation, migration, invasion and angiogenesis by negatively targeting FGF-2 in esophageal adenocarcinoma cells (EACs) and promoting NOS2 in tumor-associated macrophages (TAMs) (**A5**). Arginine availability also controls proliferation induced by NO-derived NOS2. SLC7A1 is an arginine transporter, which is negatively regulated by miR-122 (**A6**). Long non-coding RNA (lncRNAs) (HCC upregulated EZH2-associated or HEIH, urothelial carcinoma-associated 1 or UCA1, and H19) reduce miRNAs expression (miR-939-5p, miR-204 and miR-148b-3p, respectively) (**B**). miR-939-5p inhibits NOS2, which increments NO production, leading to tumor promotion (**B1**). miR-204 inhibits sirtuin 1 (SIRT1), cyclooxygenase-2 (COX-2) and NOS2 expression causing cell proliferation boost and apoptosis reduction (**B2**). miR-148b-3p upregulates NOS3 and enhances NO production leading to a negative participation in hypoxia stress and, on the other hand, this miRNA also downregulates NADPH oxidase 4 (NOX4) and increases superoxide anion production, which has a positive participation in hypoxic stress (**B3**).

## Data Availability

Not applicable.

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
