# Peer review of "Role of Nitric Oxide in Gene Expression Regulation during Cancer: Epigenetic Modifications and Non-Coding RNAs"

_ijms, 2021, doi:10.3390/ijms22126264_

Round 1

Reviewer 1 Report

The review is touching very surface of epigenetic role of NO in cancers though the target is quite interesting. I feel that they should work further on the description of the roles of NO in epigenetic regulation. Additionally, it is better to focus on epigenetic regulation but not noncoding RNA as focus of the topic is diluted (unless they make a strong and consistent link between epigenetic regulation and noncoding RNA . More, they should prepare nicer tables to summarize the role of NO in epigenetic regulation (indicating the type of cancer, type of modification of histone and DNMTs). I wrote down the notes for their information.

L68 What is NRF2? provide full name

L119 Mechanism for how NO increases DNMTs activity should be provided through expression or kinase activity? It should be detailed.

L197 The authors claim major roles of NO in epigenetic regulation, however, the provided facts are just loss of function of HDAC and CBP (not all but majority). I think the pathway to produce NO can be can be connected with histone acetylation related enzymes (which is not novel) and the author should provide the information for the mechanism how NO acts on histone acetylation and therefore the target genes (regulation).

L227 Similar is true here. Authors touch very surface of histone methylation (eg G9a, KDM3A). there are several (more than 10) histone methyltransferases and they should summarize all the relationship with NO. this should be systemically described or tabled.

L300. Noncoding RNAs are listed as one of sections. However, noncoding RNA is not direct factor to control epigenetic mechanism. If the authors want to add the noncoding RNA section, they should change the title. Only the relation between noncoding RNA and DNMT is shortly described. Additionally, the table should tell which cancer type they are presenting.

Author Response

Reviewer 1

The review is touching very surface of epigenetic role of NO in cancers though the target is quite interesting. I feel that they should work further on the description of the roles of NO in epigenetic regulation. Additionally, it is better to focus on epigenetic regulation but not noncoding RNA as focus of the topic is diluted (unless they make a strong and consistent link between epigenetic regulation and noncoding RNA. More, they should prepare nicer tables to summarize the role of NO in epigenetic regulation (indicating the type of cancer, type of modification of histone and DNMTs). I wrote down the notes for their information.

Thank you very much for your comments which are really improving the manuscript. We detailed in the following lines the modification according to your suggestions that have been included in the revised version of the manuscript.

L68 What is NRF2? provide full name

“NRF2” has been identified in the line 154. In addition, it has been incorporated in the abbreviation list

L119 Mechanism for how NO increases DNMTs activity should be provided through expression or kinase activity? It should be detailed.

We have included the described mechanism through which NO increases DNMTs activity according to the paper by Hmadcha et al. In particular, “NO induces DNMTs posttranscriptional activity increase resulting in the accumulation of CpG island methylation and suppression of gene expression” (lines 210-212, page 5).

L197 The authors claim major roles of NO in epigenetic regulation; however, the provided facts are just loss of function of HDAC and CBP (not all but majority). I think the pathway to produce NO can be connected with histone acetylation related enzymes (which is not novel) and the author should provide the information for the mechanism how NO acts on histone acetylation and therefore the target genes (regulation).

Thank you very much for your recommendation. We have extended information regarding the regulation of histone acetylation by NO. In particular, NO inhibit HDAC6 by S-nitrosation. The mechanism has been described in lines 334-341 (page 7).

L227 Similar is true here. Authors touch very surface of histone methylation (eg G9a, KDM3A). there are several (more than 10) histone methyltransferases and they should summarize all the relationship with NO. this should be systemically described or tabled.

The reviewer is correct. The section has been extended (lines 375-378 in page 8; lines 392-396, and 403-461 in pages 9-10). We hope that the value of the section has increased.

L300. Noncoding RNAs are listed as one of sections. However, noncoding RNA is not direct factor to control epigenetic mechanism. If the authors want to add the noncoding RNA section, they should change the title. Only the relation between noncoding RNA and DNMT is shortly described. Additionally, the table should tell which cancer type they are presenting.

We agree that noncoding RNA is not direct factor to control epigenetic mechanism. The reason for the incorporation of noncoding RNA to the reviewer is to enlighten the following steps of gene regulation. In addition, more connection might appear in the next future. According to the suggestion of the reviewer we have changed the title of the manuscript “Role of nitric oxide in gene expression regulation during cancer: epigenetic modifications and non-coding RNAs”. We have incorporated information regarding cancer type in Table 1.

Reviewer 2 Report

The manuscript is well written, comprehensive, it describes well the physiological role of NO in epigenetic regulation of cancer in the context of ncRNAs.

Author Response

Reviewer 2

The manuscript is well written, comprehensive, it describes well the physiological role of NO in epigenetic regulation of cancer in the context of ncRNAs.

Thank you very much for your comments. New research will appear in the next future regarding this strategic field.

Round 2

Reviewer 1 Report

The authors revised the manuscript based on the reviewers suggestions. However, for the better understanding of their work, it is recommended  to add the following information in the tables.

  1. Table 1 should have the information to describe how NO is associated with the presented noncoding RNA and lnc RNAs when they are absent and present in different cancer types. Such information summarize how NO is connected to these noncodingRNAs and lncRNAs.
  2. In the revised manuscript, they added additional information about the crosstalk between NO and HATs, HDACs, HTMs (Histone methyltransferases) and HDMs (Histone demethylase). They can summarize it in a table so that we can clearly see the association between these factors and NO in the context of epigenetic regulation.

Author Response

Jordi Muntané

Institute of Biomedicine of Seville

Laboratory 209

Hospital University “Virgen del Rocío”

Av. Manuel Siurot s/nº

41013 Seville

Spain

Phone: +34 955 923122

Mobile: +34 607 730985

Fax: +34 955 923101

e-mail: jmuntane-ibis@us.es

June 6, 2021

Prof. Dr. Maurizio Battin

Editor-in-Chief International Journal of Molecular Sciences

Dear Professor,

We are pleased to enclose our second revised version of the manuscript entitled “Role of nitric oxide in gene expression regulation during cancer: epigenetic modifications and non-coding RNAs” to be considered for publication in International Journal of Molecular Sciences. The manuscript is an invitation for the Special Issue entitled "Nitric oxide: Physiology, Pharmacology and Therapeutic Applications" edited by Prof. Khosrow Kashfi. The manuscript has been revised according to the queries of the reviewer enclosed in the following pages. The title has been changed to “Role of nitric oxide in gene expression regulation during cancer: epigenetic modifications and non-coding RNAs”. The affiliations have been indicated in english. The manuscript is not currently under consideration elsewhere and the work reported will not be submitted for publication elsewhere until a final decision has been made as to its acceptability by the Journal. The manuscript included 3 Figures and 2 Tables. The authors declare there are any competing financial interests in relation to the work described.

Thank you very much. We look forward to hearing from you. Your sincerely.

Jordi Muntané  

Reviewer 1

The authors revised the manuscript based on the reviewers’ suggestions. However, for the better understanding of their work, it is recommended to add the following information in the tables.

Thank you very much for your comments. The manuscript has greatly improved with your suggestions.

  1. Table 1 should have the information to describe how NO is associated with the presented noncoding RNA and lnc RNAs when they are absent and present in different cancer types. Such information summarize how NO is connected to these noncodingRNAs and lncRNAs.

The previous Table 1 has been renumbered to Table 2. We have incorporated the following information to the Table 2:

  • Molecular mechanism
  • Interaction with NO
  • Impact of the regulatory mechanism in carcinogenesis
  • References

  1. In the revised manuscript, they added additional information about the crosstalk between NO and HATs, HDACs, HTMs (Histone methyltransferases) and HDMs (Histone demethylase). They can summarize it in a table so that we can clearly see the association between these factors and NO in the context of epigenetic regulation.

The information has been included in the new Table 1 that incorporates information regarding:

  • Crosstalk between NO and epigenetic regulators”
  • “Impact of the regulatory mechanism in carcinogenesis”
  • “References”
